# Evaluation of Activated Carbon and Platinum Black as High-Capacitance Materials for Platinum Electrodes

**DOI:** 10.3390/s22114278

**Published:** 2022-06-03

**Authors:** Andrew Goh, David Roberts, Jesse Wainright, Narendra Bhadra, Kevin Kilgore, Niloy Bhadra, Tina Vrabec

**Affiliations:** 1Physiology Biophysics, Case Western Reserve University, Cleveland, OH 44106, USA; ajg189@case.edu (A.G.); dxr252@case.edu (D.R.); 2Chemical and Biomolecular Engineering, Case Western Reserve University, Cleveland, OH 44106, USA; jsw7@case.edu; 3Biomedical Engineering, Case Western Reserve University, Cleveland, OH 44106, USA; dr.narendra.bhadra@gmail.com; 4Physical Medicine and Rehabilitation, MetroHealth Medical Center, Case Western Reserve School of Medicine, Case Western Reserve University, Cleveland, OH 44109, USA; klk4@case.edu (K.K.); nxb26@case.edu (N.B.)

**Keywords:** nerve block, electrodes, carbon, platinum black, safety testing, performance testing

## Abstract

The application of direct current (DC) produces a rapid and reversible nerve conduction block. However, prolonged injection of charge through a smooth platinum electrode has been found to cause damage to nervous tissue. This damage can be mitigated by incorporating high-capacitance materials (HCM) (e.g., activated carbon or platinum black) into electrode designs. HCMs increase the storage charge capacity (i.e., “Q value”) of capacitive devices. However, consecutive use of these HCM electrodes degrades their surface. This paper evaluates activated carbon and platinum black (PtB) electrode designs in vitro to determine the design parameters which improve surface stability of the HCMs. Electrode designs with activated carbon and PtB concentrations were stressed using soak, bend and vibration testing to simulate destructive in vivo environments. A Q value decrease represented the decreased stability of the electrode–HCM interface. Soak test results supported the long-term Q value stabilization (mean = 44.3 days) of HCM electrodes, and both HCMs displayed unique Q value changes in response to soaking. HCM material choices, Carbon Ink volume, and application of Nafion™ affected an electrode’s ability to resist Q value degradation. These results will contribute to future developments of HCM electrodes designed for extended DC application for in vivo nerve conduction block.

## 1. Introduction

Pathological nerve activation contributes to many difficult to treat neurological symptoms (e.g., neuromuscular disorders, peripheral neuropathy, resistant hypertension, and trigeminal neuralgia) [1,2,3,4]. Clinically, nerve activation can be mitigated using several different nerve blocking techniques such as nerve ablation, neurotomy, opioids, and local analgesic blocks. These methods are not always effective and have several limitations: nerve ablation and neurotomy are irreversible, and opioid use can lead to addiction, overdose, and side effects such as nausea and liver damage. Medication dosage can be difficult to adjust and titrate for each patient. Local analgesics are not immediately effective or reversible and provide only short-term relief, requiring treatment every 3 months. Electric nerve block is capable of overcoming many of the adverse effects associated with these current methods.

Direct current (DC) can be used to generate a rapid and reversible nerve conduction block suited to treat pathological nerve activity [5,6,7,8,9,10,11]. DC produces nerve block by depolarization, which inactivates sodium channels preventing the propagation of action potentials. However, prolonged DC application has previously been demonstrated as unsafe in both acute and chronic settings [12,13,14,15]. Permanent nerve conduction loss is likely a result of non-reversible faradic reactions such as oxygen reduction that generate reactive species [16,17,18]. Faradic reactions occur when the charge applied to the electrode cause it to operate above its capacitive region. These reactive species may accumulate at the electrode-nerve interface and damage tissues. Therefore, for DC nerve conduction block to be safe and reliable, the electrode design must minimize the production of reactive species. This can be accomplished with appropriate electrode surface chemistry such as high-capacitance materials (HCMs).

Charge balanced direct current (CBDC) waveform designs and integration of HCMs in electrode designs facilitate the injection of DC with minimal reactive species production [19,20]. The HCMs expand the capacitive region of the electrode to allow for a larger application of charge before reactive species are produced allowing DC to be applied for a longer period of time and at higher amplitudes. This CBDC waveform is considered DC (instead of alternating current) because its timeframe is such that the response of the nerve is as if it were a DC waveform. To generate charge-balanced direct current, the waveform shape includes a phase that is equal but opposite to the charge of the initial phase [12]. This translates to a waveform with a cathodic block phase and an anodic recharge phase [21,22]. The cathodic phase waveform of a charge balanced direct current contains a plateau phase which generates the block, with a ramp-to-plateau and ramp-to-recharge phases to avoid neural activation (i.e., onset response) [11]. Two types of HCMs that increase the capacitive ability of conductive devices are PtB and activated carbon [23,24,25,26,27,28,29,30,31]. The incorporation of these two HCMs allows for the fabrication of electrodes that have a large enough capacitive/pseudocapacitive region to safely apply DC block for seconds to minutes before charge balancing is necessary [21,22]. Platinum black electrodes are fabricated by electroplating platinum powder onto the contact surface of the electrode [32,33]. This process increases the effective surface area of an electrode, thus increasing the capacitance/psuedocapacitance region and avoiding the production of reactive species [12]. Activated carbon electrodes are fabricated by creating carbon ink (CI) and applying the CI onto the electrode contact surface. This creates a capacitive layer on top of the electrode surface, thus increasing the electrode’s capacitance and reducing reactive species production. In this paper, variations of these two HCM electrode designs are evaluated on whether they can perform on the timescale and capacitance necessary for DC block techniques [21,22,34].

The “Q” value is the metric that specifies the amount of charge that can be delivered before reactants are produced at the electrode surface. A cyclic voltammogram (CV) is performed on each electrode and the total charge of the capacitive/psuedocapacitive region is calculated [22]. The “Q” value estimates the capacitive storage charge limit of an electrode, which aids in avoiding tissue damage when applying DC charge [22,35,36]. HCM electrodes with a sufficiently large Q value and utilizing a CBDC waveform can provide safe DC delivery for peripheral nerve conduction block [22,37]. However, HCM electrodes have demonstrated Q value degradation and irreversible nerve conduction loss with prolonged DC application [22,38]. Therefore, the purpose of this study is to characterize the Q value degradation of HCM electrodes to determine a design for chronic DC peripheral nerve block application that avoids reactive species production as a result of Q value degradation.

## 2. Materials and Methods

### 2.1. Monopolar Electrode Design

This study utilizes a monopolar J-cuff electrode design that has demonstrated robust and repeatable recruitment of nerve fibers during stimulation and conduction block [39,40,41]. A more detailed description of the design and fabrication process is described in Foldes 2011 [42] and Vrabec 2016 [22]. The fabrication begins with a 25 µm-thick platinum sheet being cut into a 4 × 6 mm rectangle and welded to a platinum wire. Both the platinum sheet and lead wire are then sheathed in silicon. A 2 × 3 mm window is cut from the silicon sheeting over the platinum sheet which becomes the interface to the external environment. The electrodes are then electrochemically cleaned using a square wave of +3 mA for 10 s, followed by −3 mA for 10 s, for 25 cycles. This cleaning was performed in 0.1 M H_2_SO_4_. Cleaned electrodes were rinsed and dried and then stored in closed Petri dishes until high-capacitance material was applied. 

### 2.2. Cyclic Voltammogram

At key points in the study, CVs were generated for each HCM electrode to determine their “Q” value. The CVs were generated using a Solartron Inc. (Farnborough, UK)., Model 1280B Potentiostat, using a BASi (West Lafayette, IN, USA) RE5B reference electrode (measurements are accurate to within 10 mV) with a sweep rate 10 mV/s, voltage range of −0.255 to +1.20 V, and sampled at 10 Hz. Both PtB and CI required different methods of CV analysis. The Q value was estimated for PtB electrodes by calculating the pseudocapacitive charge associated with hydrogen adsorption/desorption by integrating the current in each of these regions and calculating the average area of the two regions. The CI electrodes’ Q value represents the capacitive region and is estimated by integrating the charge/discharge regions of the CV and calculating the average area of the two regions. Figure 1 and Figure 2 demonstrate obtaining the Q value from the CVs of CI and PtB high-capacitance electrodes.

### 2.3. Activated CI Electrodes

CI was made using the following materials: YP-50 (activated carbon powder obtained from Kuraray Chemicals, Chiyoda City, Tokyo, Japan), n-methyl pyrrolidone (NMP, Fischer Scientific: N1401, Hampton, NH, USA), and polyvinylidene fluoride powder (PVDF, Millipore Sigma 2937-79-9, Burlington, MA, USA). The process to create CI is as follows:Prepare binder solution by dissolving 1.0 g of PVDF in 9.0 g of NMP and stir with magnetic stir bar until PVDF powder is fully dissolved;Combine 3.0 g of YP-50 and 6.0 g of NMP into a vial with magnetic stirrer;Add 3.0 g of binder solution dropwise to YP-50 and NMP dispersion (created in step 2);Stir resulting mixture for 2 h with magnetic stir bar.

This CI formula was varied in our experiments to create two additional CI formulas (+60% with 4.8 g YP-50 and −60% binder with 1.2 g YP-50). Either 4 μL or 8 μL of CI was then micro-pipetted (scilogex MicroPette, Rocky Hill, CT, USA, 0.5–10 μL) onto the platinum window of the electrode. This results in either a 0.7 mm or a 1.4 mm film thickness. The ink was applied under a microscope and a 1mm diameter rod was used to even out applications of ink and disperse air bubbles. After application, the entire electrode was placed in an oven set at 200 °F (93.3 °C) for 30 min. The electrode was then soaked in Ringers lactate solution for 30 min before any measurements or manipulations.

### 2.4. CI Binder Concentration Study

Fifteen 3 × 2 mm platinum electrodes were fabricated. CI was created with five varying binder concentrations: +60%, +30%, +0%, −30%, and −60%. Each ink formula was applied to three different electrodes at volumes of 10 μL, 12.5 μL, and 15 μL. Q values were tested after CI application.

### 2.5. Platinum Black Electrodes

Electrodes were electrochemically treated to create a platinum black surface [24]. Platinum black was deposited from a chloroplatinic acid solution (5 g H_2_PtCl_6_ in 500 mL H_2_O, with NaCl (2.9 g) and lead acetate (0.3 g)). The setup for electrode platinization is as follows: A glass vessel was filled halfway with platinizing solution. The electrode being platinized was secured with its window facing a platinum counter electrode (max distance 5 cm). The glass vessel was then suspended in an Elma Ultrasonic Bath that was filled with DI water. A galvanic square wave (−56 mA/cm^2^ “on” current for 5 s, followed by 5 s at open circuit = 10 s/cycle) was applied with a potentiostat/galvanostat (Solartron Analytical 1280B Electrochemical Test System). Sonication occurred continuously during the galvanic square wave cycle. The potentiostat/galvanostat was interfaced with a computer using a USB compatible GPIB Instrument Control Device. Galvanic wave parameters were set with CorrWare (Scribner Associates) software. The Q value of the high-capacitance electrode was controlled by the number of deposition cycles set in CorrWare. The average number PtB electroplating cycles needed to reach ~20 mC for new electrodes was 400–800 cycles. Fewer cycles were needed for previously plated (re-used) electrodes.

### 2.6. Nafion™ (Sulfonated Tetrafluoroethylene Based Fluoropolymer-Copolymer)

Nafion™ has been suggested to increase the adhesion of conducting composites found in supercapacitors [43,44,45]. We hypothesized that Nafion™ would increase the stability of our HCM electrode designs by preventing the coating from peeling off. After HCM application, 4.0 μL (0.66 μL/mm^2^) of Nafion™ (5 wt.% in isopropyl alcohol) was micropipetted onto the electrode window and then secured by heating in a conventional oven at 90 °C for 30 min. Nafion™ was only added to the terminal trial of each electrode design (5th trial out of 5 total trials) to avoid affecting other trials with residual Nafion™.

### 2.7. Testing Q Value Resilience (Soak, Vibrate, and Bend Tests)

A battery of tests was developed to measure the resilience of the “Q” value of high-capacitance electrodes. The battery consisted of subjecting high-capacitance electrodes to soak, vibrate, and bend testing. These tests were designed to stress the adhesion of the high-capacitance material. Although there are other metrics for evaluating the adhesion of the surface, Q value was chosen as a functional metric that would also measure any fouling of the material that would not be apparent from visual inspection. Platinum black and CI was applied to 27 monopolar electrodes (electrode specifications in Section 2.1). The variations in HCM electrode design of CI and PtB are explained in their respective sections. New formulations of carbon ink were created for each separate test (soak, vibrate, bend).

#### 2.7.1. Soak Test

The soak test consisted of 12 electrodes with variable HCM designs soaked for 70 days. Six CI electrodes consisting of two electrodes of each binder concentration (regular, +60%, and −60%) one with Nafion™ and one without. Six PtB electrodes were used, consisting of four electrodes with a Q value above 20 mC (“high”) and two electrodes with Q < 20 mC (“low”). Two of the high value electrodes and one of the low value electrodes were also coated with Nafion™. Soaking consisted of the windowed end of each electrode being submerged in a 4 mL glass vial filled with lactated ringers. Electrodes were grouped in vials by CI binder concentrations (regular, plus, and minus) and platinum black initial Q value (high, medium, and low). The Ringers solution was replaced every two weeks to prevent any organic growth in the media. Q value measurements through CVs were performed weekly until the 50–60 day mark. Soaking continued after the 50–60 day mark with less frequent “Q” value measurements.

#### 2.7.2. Vibrate and Bend Tests

Both vibrate test and bend test shared a similar design with 7 electrodes each. Six CI electrodes were assigned one of three binder concentrations (regular, +60%, and −60%) and one of two CI volumes (4 μL and 8 μL). Only one PtB electrode was assigned for each test. Each electrode underwent consecutive trials of vibrate/bend until its Q value fell below 10 mC, followed by recoating, initial Q value CV, then undergoing its next trial of vibrate/bend testing. Nafion™ was added to the fifth (final) trial of every set to avoid the contamination of any subsequent trials. Vibration trials consisted of immersing the electrodes in an Elma Ultrasonic Cleaner filled with DI water. The electrode window end was submerged 2–3 inches into the bath. The ultrasonic bath’s vibration function was turned on for 1–5 min and then the Q value of the electrode was tested. The vibration interval started at 1 minute and was increased to 5 min if the electrode’s Q value did not fall below 10 mC within ten cumulative minutes of vibration.

Bend test electrode windows were curved around a 1 mm stiff metal rod to simulate J-cuff shaping in preparation for nerve placement [39,42]. A single bend trial consisted of first slowly bending and then unbending the electrode window. After the unbend stage, the electrode underwent Q value testing. A trial continued as long as the electrode’s Q value did not fall below 10 mC.

### 2.8. Statistical Analysis

Normality of data was established using the Shapiro–Wilk W Test (JMP 15). A one-way ANOVA was used to evaluate the effects of selected independent variables if a dataset’s normality was supported by this test or if its sample size was greater than thirty. The one-way ANOVA is robust against violations of the normality assumption [46]. Non-parametric testing (e.g., Kruskal–Wallis test) was employed in data sets where normality could not be supported, and the sample size was less than 30 samples. Comparison of means among data sets with statistically significant results and multiple groups was performed by the Student’s t test. All independent variables were included in the model to determine the effect of each of the variables on the model. Fit lines were evaluated for each data set.

The Q value of CI electrodes from the binder concentration study were evaluated using CI binder concentrations (−60%, −30%, regular, +30%, +60%) and CI volume (10 μL, 12.5 μL, and 15 μL) as independent variables.

For the initial Q value of electrodes, the independent variables included were: HCM material (PtB and CI concentrations: −60%, regular, +60%), volume (4 μL and 8 μL), and Nafion™. The sample size included all electrodes fabricated for soak, vibrate, and bend testing.

For the soak testing, the soaking time (days) to failure was evaluated by the independent variables of HCM material (PtB and CI binder concentrations: −60%, regular, +60%) and Nafion^TM^. Failure threshold was set at 90% of each electrode’s stabilized value. Additionally, the initial Q value changes in response to soaking were analyzed as “amount stabilized” (|Initial Q − Stabilized Q|).

For the vibrate and bend testing, both vibrate time (minutes) and bends until failure were evaluated by the independent variables: HCM material (PtB and CI concentrations: −60%, regular, +60%) and CI volume (4 μL and 8 μL), Nafion™, and the electrodes’ 1st trial time to failure vs. trials 2–5 time to failure. Vibrate failure threshold was set at 90% of initial Q value. Bend failure threshold was set at 70% of initial Q value (see Section 4).

For the amount stabilized (derived from stabilization value—initial value) of each electrode, the independent variables were HCM material (All PtB and CI concentrations stated above), volume (4 μL and 8 μL), and Nafion™. The sample size included all electrodes fabricated for soak, vibrate, and bend testing.

JMP’s life distribution platform was utilized to model the lifespan of electrodes that underwent vibrate and bend tests. The analysis predicts the probability of an event occurring based on a time to event variable. Probability models were created for vibrate and bend tests in order to predict failure based on time vibrated or number of bends. Shaded areas represent the confidence intervals.

## 3. Results

### 3.1. CI Binder Concentration Study

Figure 3 visualizes the results of Q value vs. concentration of activated carbon powder (YP-50) by groups of differing volumes of CI. The Shapiro–Wilk test supported normality (*p* > 0.05) of the data set. A one-way ANOVA found a significant effect (*p* = 0.0076) of CI concentration on Q value irrelevant of volume. No significant effect was found from volume on Q value per a one-way ANOVA (*p* > 0.05). Trend lines were applied to the data points revealing an upward trend from increased powder concentration.

### 3.2. High-Capacitance Material Electrodes Initial Q Value

Initial Q values were obtained from CVs immediately after application of HCM (CI-minus 60% binder, CI-regular binder, CI-plus 60% binder, and platinum black). The means of initial Q values are listed in Table 1. The Shapiro–Wilk test was not performed because the sample size was greater than thirty. A one-way ANOVA found a statistically significant effect of HCMs on the initial Q value (*p* = 0.0013). A comparison of means by a Student’s t test on the HCM group found significant differences between PtB vs. CI-regular (*p* = 0.0027), PtB s vs. CI-minus (*p* = 0.0001), and PtB vs. CI-plus (*p* = 0.0039). There were no significant differences between CI groups (i.e., CI-regular vs. CI-minus, CI-regular vs. CI-plus, CI-minus vs. CI-plus). There was no statistically significant effect of CI volume, repeated use of same electrode, or Nafion™ on initial Q value (*p* > 0.05) per one-way ANOVA. Figure 4 shows a box plot of the relative initial Q value for each type of HCM.

### 3.3. Soak Test

Both CI and PtB displayed unique Q value changes in response to initial soaking. CI electrodes demonstrated an increase in Q value as quickly as two days after soaking (Figure 5), and by day 16, the Q values had risen to an average of 130% (Std Error = 7.8%) of their initial Q value. This rise in Q value was followed by a plateau where their Q values decreased or increased minimally. The plateau phase was followed by a rapid failure phase where Q values suddenly dropped significantly below the failure threshold. Failure threshold was reached at day 45 on average.

PtB trials demonstrated an initial decrease in Q value with the majority of Q value decrease occurring in the period of days 2–8 of soaking (Figure 6). During this initial decrease phase, Q values decreased to an average of 74% (Std Error = 7.6%) of their initial Q value. After this initial decrease phase, the Q values of PtB electrodes stabilized into a plateau phase. In contrast to CI electrodes, the Q values of PtB electrodes decreased gradually to the failure threshold. Failure threshold was reached at day 36 on average. Because of the consistency of these changes, we defined relative changes in the Q value in terms of the Q value after these changes stabilized.

The use of the stabilized Q value allowed us to normalize the Q values to compare electrodes with different Q values. Additionally, it allowed us to define the failure threshold at a functional level. For the PtB, all electrodes would have reached the failure threshold before the stabilization, thus making comparisons between techniques infeasible. To obtain the stabilized Q value, we first filtered the Q values from each soak trial with a moving average (interval = 3). With these filtered Q values, a slope [(Q2 − Q1)/Day2 − Day1)] vs. days soaked graph created in JMP. Stabilization values were obtained from this graph if they met our stabilization criteria. For CI electrodes, an electrode was considered stable if the rate of change in its Q value remained below 2 mC/day for 10 days. Similarly, a PtB electrode was considered stable if its Q value remained below 0.2 mC/Day and stayed below this value for 10 consecutive days. The days soaking until stabilization and their respective Q value were elucidated, and each trial’s failure threshold was calculated using this Q value. Failure threshold was defined as 90% of a trial’s stabilized value. The use of this stabilized Q value blunted the significant effect of initial Q value changes in failure threshold. The amount stabilized was evaluated separately with CI and PtB groups. A one-way ANOVA found no significant effect of initial Q, CI volume, Nafion™, or CI binder concentrations on amount stabilized.

The Shapiro–Wilk test did not support normality (*p* < 0.05) of the soak data set. Both one-way ANOVA and Kruskal–Wallis test were performed to evaluate the independent variables relationship with days soaking until failure. The one-way ANOVA found a significant effect of Nafion™ on days soaking until failure (*p* < 0.05). Electrodes with Nafion™ maintained Q values above failure threshold for 57.2 days on average (Std Error of Mean = 5.0) and electrodes without Nafion™ maintained Q values above failure threshold for 31.3 days on average (Std Error of Mean = 10.2). Both the one-way ANOVA and Kruskal–Wallis test found no significant effect of HCM (CI-minus, CI-plus, CI-regular, PtB low, PtB medium, and PtB high) on days soaking until failure (*p* > 0.05).

### 3.4. Vibrate Test

The Shapiro–Wilk test did not support normality (*p* < 0.05) of the data set. A one-way ANOVA found significant effects of both HCM material (*p* = 0.0256) and CI volume (*p* = 0.0489) on vibration time until failure (90% Failure threshold). Using a *p* < 0.05 criteria for significance, a comparison of means by a Student’s t test on HCM groups found significant differences between PtB vs. CI-regular (*p* = 0.0148) and CI-plus vs. CI-regular (*p* = 0.0161). There were no significant differences between CI groups: PtB vs. CI-minus, PtB vs. CI-plus, CI-regular vs. CI-minus, and CI-minus vs. CI-plus (*p* > 0.05). CI-regular electrodes failed significantly sooner than PtB and CI plus. If an electrode had a CI volume of 4 μL it failed significantly failed sooner than an electrode with a CI volume of 8 μL. Statistical analysis by one-way ANOVA and Kruskal–Wallis test found no significant statistical effect of Nafion™ or 1st trial vs. trials 2–5 on vibration time to failure (*p* > 0.05). HCM and CI volume life distribution analysis are contained in Figure 7 and Figure 8.

### 3.5. Bend Test Results

The Shapiro–Wilk test did not support normality (*p* < 0.05) of the data set. A Kruskal–Wallis Test found a significant effect of HCM groups on the number of bends until failure (*p* = 0.0360). Comparison of means by Student’s t test for HCM groups found significant differences between CI-minus vs. PtB (*p* = 0.0339) and CI-minus vs. CI-plus (*p* = 0.0204). Statistical analysis by one-way ANOVA and Kruskal-Wallis test found no significant effect of CI volume, Nafion™, or 1st trial vs. trials 2–5 on bends until failure (*p* > 0.05). Figure 9 is a life distribution analysis between HCM groups.

## 4. Discussion

The fabricating, testing, and failing of the electrodes provided key insights on HCM electrodes and their Q value resilience to environmental factors. One of the first steps in HCM electrode work is fabricating an electrode with a high enough Q value for a planned waveform. The initial Q value of a HCM electrode is significantly affected by choosing either PtB or CI as the HCM. Even though PtB electrode Q values were artificially limited to create groups with differing Q values, there was difficulty creating a PtB electrode with a Q value greater than ~25 mC. The difficulty arose from diminishing Q value gains when additional electroplating cycles were completed at around the 25 mC mark. Eventually, a PtB electrode’s Q value would not increase with the addition of more electroplating cycles. This effect may be caused by decreasing adhesion of PtB on the electrode surface as the amount of electroplated PtB increases. Because we electroplated these electrodes under vibration, it is possible ~25 mC is close to the limit of PtB that will adhere to our platinum contact surface (2 mm × 3 mm).

When analyzing the effect of CI binder concentration and CI volume on initial Q value we found conflicting results. The binder concentration study (Section 3.1) supported an increase in Q value when CI binder concentration was decreased (higher concentration of activated carbon). While there appears to be a positive correlation with CI volume and Q value, our data showed no significant effect of CI volume on Q value. This lack of significant effect may be due to insufficient data points for this experiment. Additionally, Figure 3 displays a possible trend where differences in CI volume are more pronounced at lower binder concentrations (higher concentrations of YP-50). The effect of CI volume is seen in Figure 3’s lines of best fit. The slope of these lines increases with higher volumes of CI, supporting the idea that larger Q values are correlated with increased CI volume. A lack of statistical evidence may be due to the binder concentration study’s small sample size. In contrast to the binder concentration study, an analysis of the electrodes used for the soak, vibrate, and bend tests found no statistically significant effect of CI binder concentration or volume on initial Q value. Despite a lack of statistical effect, the means of CI binder concentrations and volume groups suggest an effect on initial Q value. It is possible that the range of binder concentrations (±60%) and volume (4 μL/8 μL) was not large enough to obtain significant results. While the range of CI volume is difficult to increase, due to limited space on the electrode window, the limits of binder concentration have not been tested. Therefore, future studies could increase the range of these two factors to amplify the effect on the Q value of HCM electrodes.

While CI electrodes increased in Q value upon soaking, PtB electrodes decreased in Q value. Both changes were quantified and defined as time to stabilization (Section 3.3). These differences between PtB and CI may be explained by how each interacts with the soaking environment. It may take an extended soaking time for electrolyte to fully penetrate the fine pores (sub-micron pore diameters) of CI electrodes [47]. The PtB electrode’s initial decrease may be caused by adsorption of organic contaminants, as PtB is very sensitive to organic impurities [48]. Further CV work may be needed to measure the electrolyte saturation of CI or detect organic impurities in PtB electrodes. Nafion™ demonstrated its effectiveness as a Q value protector by raising the number of days of soaking until failure. As mentioned in Section 2.4, Nafion™ contributes to the adhesion of supercapacitors. Further research could explore the volume of Nafion™ applied and its application methods. Finally, a significant statistical difference was not found between days soaking until failure and the two HCMs (CI vs. PtB). Because PtB soak testing ended before three of the PtB electrodes failed, the full extent of PtB’s resiliency to soaking is untested. Future experiments with longer timescales are needed to test PtB’s full long-term viability.

The time to stabilization helped us establish the timeframe of the initial Q value changes in response to soaking. The rate of change for PtB electrodes was much smaller, as represented by their stabilization criteria being lower than CI electrodes. The criteria for CI electrodes were larger because they displayed larger fluctuations in their Q value/day rates during this timeframe. This difference in Q value fluctuation is highlighted by the standard deviations of CI and PtB electrodes (without a moving average applied) 12.72 mC and 1.77 mC, respectively. CI electrode Q values were larger on average than PtB electrodes, which may partly explain the larger Q value/day changes. Further studies on time to stabilization and change in Q value/day may lead to a better understanding of the nature of CI’s saturation that increases Q value or PtB’s interaction with Q value decreasing material. In vibrate and bend testing, PtB and CI-plus electrode groups maintained Q values above the failure threshold for the largest amounts of time. Using life distribution analysis, we found PtB and CI-plus electrodes had an almost 30% decrease in the probability of failure when compared to the next best performing electrode (CI-minus). Surprisingly, CI-regular’s times to failure were not between CI-minus and CI-plus but instead performed worse than both other CI groups. Additionally, while CI-regular performed worse than CI-minus, the results were not statistically significant. CI-regular was found to have a 100% probability of failure at 17 min of vibrating. In comparison CI-plus and CI-minus electrodes had a sub 40% probability of failure at 17 min. Variabilities in electrode and CI fabrication may have caused this underperformance. Future studies should establish “CI-regular” formulations as an intermediary CI design that lies between high initial Q value (CI-minus) and Q value resiliency (CI-Plus).

Vibration testing results also support electrodes with CI volume of 8 μL versus 4 μL. Larger deposits of CI adhered to the electrode window surface for longer periods of time. While this concept is useful in future CI electrode designs, one can only increase CI volume to a maximum until the deposit geometrically interferes with the electrode –nerve interface. Bend test results demonstrated early failure for all HCMs when using a failure criterion of 90% and 80% of initial Q value. Most trials failed after two bends with these criteria. In order to compare groups, we used a lower failure threshold (70%) and found statistically significant results between HCM electrode designs. However, the results do not demonstrate a clear best performing electrode design. Because the best performing electrode failed within roughly 20 bends, an investigation into other electrode fabrication methods is warranted. Some possibilities include utilizing other binders for CI or replacing the Pt foil substrate to achieve significantly higher levels of adhesion between CI and the substrate. The high failure rate of the bend tests confirms that when handling HCM electrodes it is crucial to minimize the number of times the electrode window/HCM interface is bent. Additional precautions such as pre-bending electrodes before applying HCM or applying HCM while electrodes are in a partial J-Cuff position may be taken to protect against early failure secondary to bends. Although a failure criterion of 90% or 80% would be ideal, it may be that a 70% reduction in Q value still results in a clinically useful waveform. It is unlikely that in a clinical application, the electrode would be bent more than once. However, the lower failure criteria allowed us to compare the electrode fabrication methods to help us improve future designs.

## 5. Conclusions

Performing this battery of tests on HCM electrodes supports design choices to decrease Q value degradation. The inclusion of Nafion™ proved important in preserving the Q value in soak testing, CIPlus and PtB electrodes were most likely to survive bends and vibrations, and a larger CI volume resisted Q value degradation in response to vibrating. These results support the continued effort to improve HCM electrode designs in extended DC application environments to further explore the electrophysiological relationship.

## Figures and Tables

**Figure 1 sensors-22-04278-f001:**
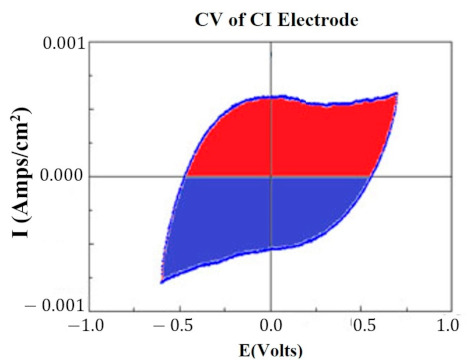
CV of CI electrode. Image is taken from CorrView (Scribner Associates, Southern Pines, NC, USA). The CV is graphed as I (current) vs. E (voltage). The Q value is an estimate of the average of the area of the red region and the area of the blue region.

**Figure 2 sensors-22-04278-f002:**
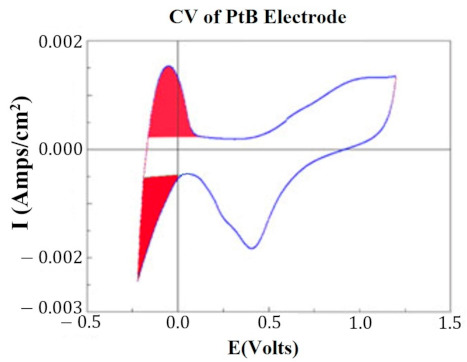
CV of platinum black electrode. Image is taken from the program CorrView (Scribner Associates). The CV is graphed as I (current) vs. E (voltage). The areas under the curve designated in red are averaged together to determine the appropriate Q value. Charge is determined by integrating over this region.

**Figure 3 sensors-22-04278-f003:**
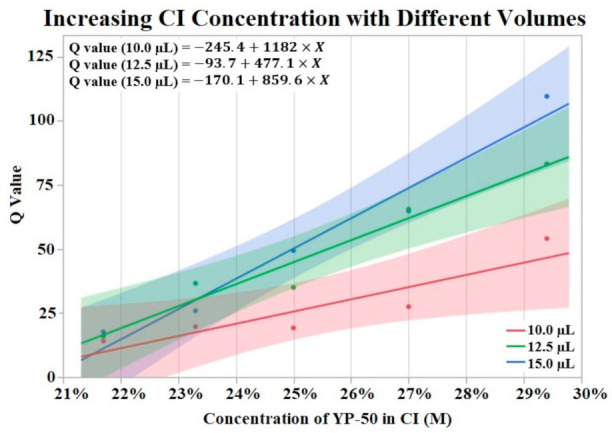
Fifteen CI electrodes were fabricated at different CI concentrations and volumes (indicated as red, green and blue). The molarity of activated carbon was calculated for each electrode concentration. An upward trend in Q value is shown for increases in concentration. α = 0.05.

**Figure 4 sensors-22-04278-f004:**
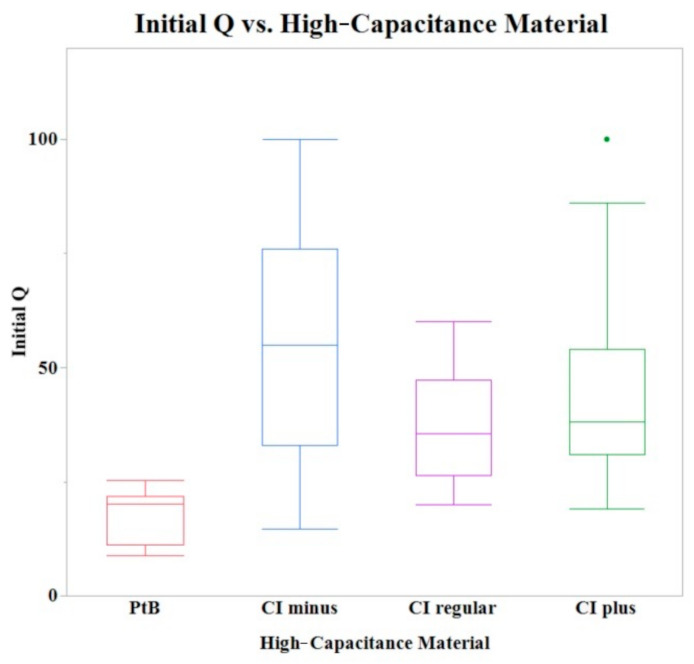
A box plot of the initial Q values for all tested electrodes. A one-way ANOVA found no statistically significant differences between CI binder concentrations (plus, minus, and regular) when PtB electrodes were omitted.

**Figure 5 sensors-22-04278-f005:**
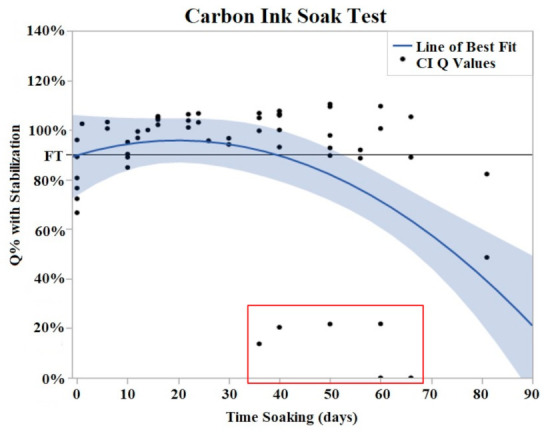
CI soak trials were plotted as Q% with stabilized value vs. time soaking (days). Stabilized values were derived from initial Q value analysis. A quadratic line of best fit was fit to the data (Y = 0.9078 + 0.004776x − 0.000131 × 2). The horizontal line at 90% represents the failure threshold. α = 0.05. The data points, indicated by the red box, demonstrated two electrodes with premature and abrupt Q value decreases.

**Figure 6 sensors-22-04278-f006:**
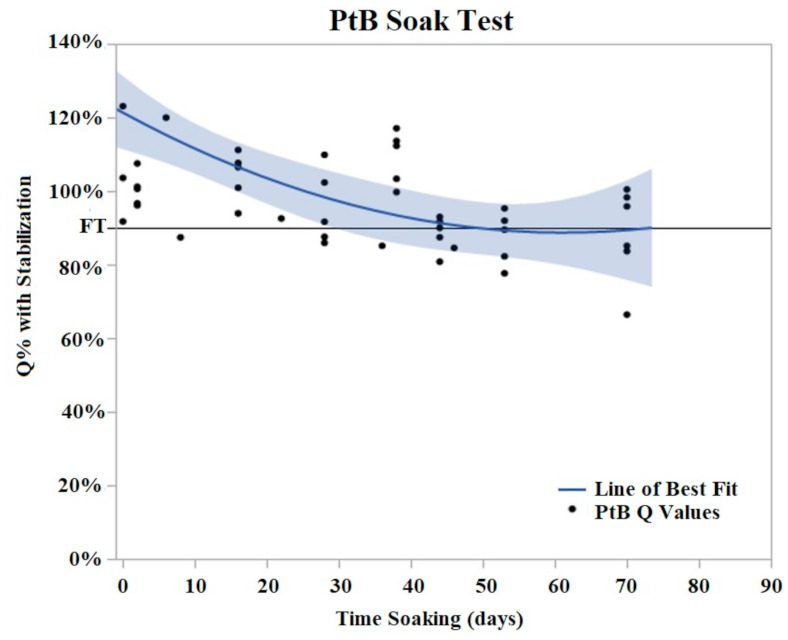
PtB soak trials were plotted as Q% with stabilized vs. time soaking (days) in. Stabilized values were derived from initial Q value analysis. A quadratic line of best fit was fit to the data (Y = 0.8979 + 0.00605x − 0.000152 × 2). The horizontal line at 90% represents the failure threshold. α = 0.05.

**Figure 7 sensors-22-04278-f007:**
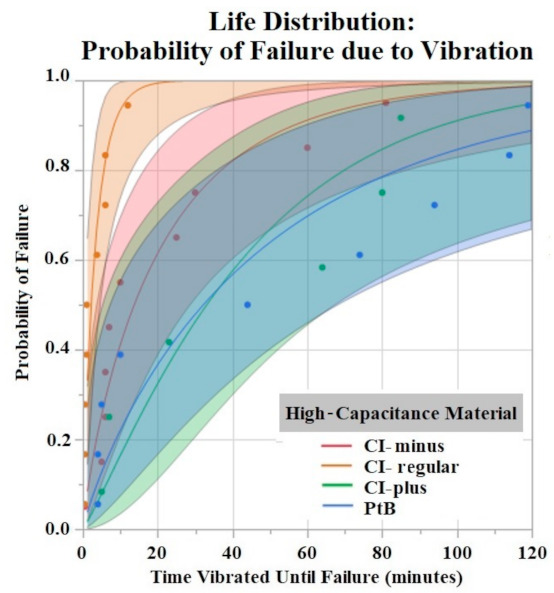
Weibull life distribution analysis of HCM groups. Binder concentrations for CI-Plus and CI-Minus are +60% and −60% binder. The shaded area represents the confidence interval of each analysis.

**Figure 8 sensors-22-04278-f008:**
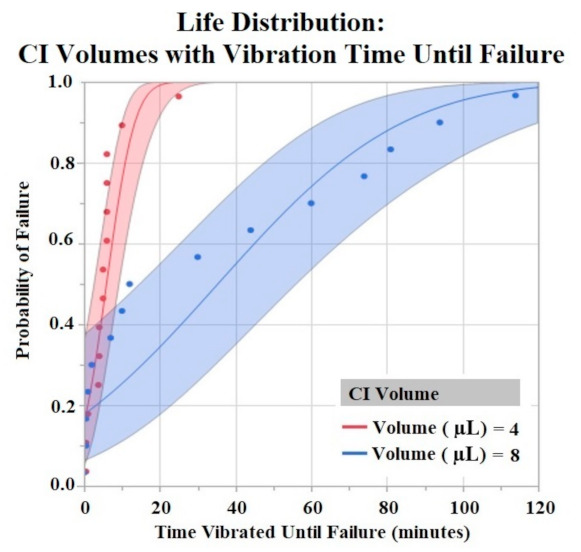
CI electrodes were separated by their CI volume (4 μL and 8 μL). Life distribution analysis produced a probability of failure model. Binder concentrations for CI-Plus and CI-Minus are +60% and −60% binder. Confidence intervals represented by shaded area. α = 0.05.

**Figure 9 sensors-22-04278-f009:**
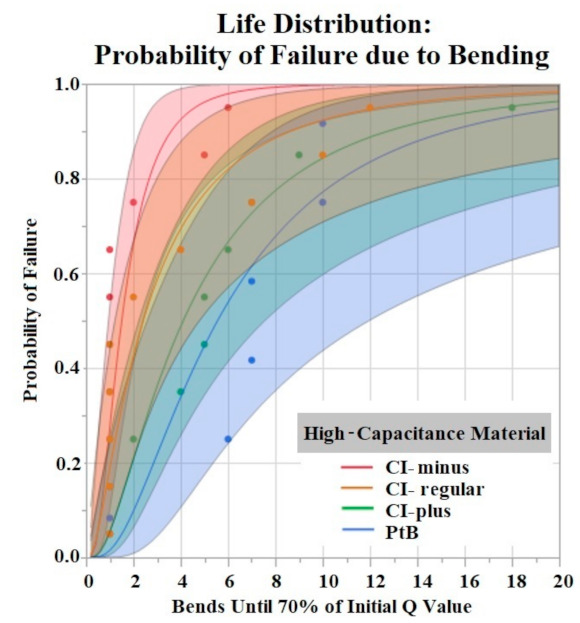
Life distribution analysis. Shaded areas indicate respective confidence interval (CI = 0.95). Distribution lognormal.

**Table 1 sensors-22-04278-t001:** Contains the four types of HCM electrodes with the number of electrodes tested (N), mean of initial Q values (Mean of Initial Q), and the standard error (SE).

HCM	N	Mean of Initial Q	SE
CI-Minus	23	53.71	5.72
CI-Regular	20	46.4	6.14
CI-Plus	21	44.96	5.99
PtB	16	17.85	6.14

## Data Availability

The data presented in this study are available on request from the corresponding author.

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
