# Peer review of "Evaluation of Activated Carbon and Platinum Black as High-Capacitance Materials for Platinum Electrodes"

_sensors, 2022, doi:10.3390/s22114278_

Round 1

Reviewer 1 Report

In this report the authors studied the stability of electrodes having high capacitance materials and the ionomer Nafion. The methods included the calculation of the Q-value in terms of some design parameters such as the volume of the HCM (for carbon ink, CI) or binder amount (for carbon ink), and in stress conditions such as soaking, vibration and bending. The authors found that Nafion preserve Q-values (although they did not include a comparative table to highlight this observation), while CI having certain amounts of binder and CI volume and PtB electrodes are more resistive to stressing conditions.

I think that the paper is well written although there is not enough information in the introduction according to my appreciation: just the very specialized reader will understand the paper in its own without the need to read other documents explaining supercapacitors features and the Charge balanced direct current protocol. I then suggest to include more info in the introduction if the authors want to facilitate the readers comprehension.

Author Response

The authors would like to thank the reviewers for their comprehensive and insightful review.  We are confident that responding to these comments will improve the quality and applicability of our paper.  We appreciate the opportunity to improve the clarity of our research.

  1. Additional content has been added to the introduction to further describe the application and the electrochemistry.

Reviewer 2 Report

The work concerns the synthesis and analysis of activated carbon and platinum black electrodes as high capacitance materials. My comments to the work are presented below:

  1. There are no references in the bibliography to topics related only to activated carbon electrodes.
  2. I suggest you put more CVs to show how the curve changes depending on the sample type.
  3. I suggest determining such basic data as the size of the specific surface or the thickness of the coatings.
  4. In line 131, the values 4.8 g and 1.2 g are incomprehensible, besides the + 60% and -60% markings are insufficiently explained. There is no consistency in naming samples, once is "-60%", once is "minus60".
  5. On line 191 there is a sentence: "The Ringers solution was replaced every two weeks or if any organic growth was observed." How many times has this behavior been observed and when? Has any relationship been observed?
  6. Data from section 3.2 will be better visible in the form of a table than in the text.
  7. In figure 5, the CI is not shown depending on the type of electrodes. Why?
  8. I found "significant differences" in section 3.4 without any details, please specify.
  9. There is a mention of mechanical stability in line 409 - how was this observed?

Author Response

The authors would like to thank the reviewers for their comprehensive and insightful review.  We are confident that responding to these comments will improve the quality and applicability of our paper.  We appreciate the opportunity to improve the clarity of our research.

  1. Additional references have been added
  2. The CVs that are included in the paper are primarily used to demonstrate how the Q value was calculated. Although there is some variation in the CV shape, the Q value represents the magnitude of the capacitance.  The clearest difference in shape is between the Platinum Black and the Carbon electrodes due to the material difference. Since hundreds of CVs were generated for this paper, we feel that the Q value is a clearer metric for representing capacitance than the shape of the CV.
  3. The size of the electrode surface is described in section 2.1 “Monopolar Electrode Design” Although we didn’t measure the thickness specifically, the size of the electrode surface and the volume of carbon ink has been used to estimate the thickness of the coating for the carbon electrodes and this information has been added to the paper.
  4. The definition of the carbon ink fabrication has been clarified and the inconsistencies in the labelling has been corrected.
  5. The observation of organic growth occurred early in our testing. The 2 week replacement protocol prevented further growth.  The description has been simplified to clarify this.
  6. A table now replaces the text.
  7. The intent of the soak test was to determine the effect of soaking on the Q value for CI and PtBlack. So, the Q values for each type of electrode was consolidated on a single diagram in order to show the trends.  Also, in the text there was a typographical error referring to figure 5 for the Pt Black data instead of figure 6.  This has been corrected.
  8. “Significant differences” refers to statistical significance. The significance criteria has been added for clarification.
  9. “Mechanical stability” has been defined in section 2.4 when describing Nafion.

Reviewer 3 Report

To mitigate the damage caused by prolonged DC on  nervous tissues,  the design of high capacitance materials (HCMs), i.e., activated carbon and platinum black in this manuscript,  are evaluated. Different kinds of components in HCMs are prepared and carefully examined, the results are analyzed according to the metric of Q value. Generally, the experimental descriptions demonstrated this work may have practical value in improving electrophysiological sensors, however, the following issues are still necessary to be addressed before it can be accepted.

1) Format should be carefully checked.

2) only one high Q value electrode is not enough ( as mentioned in section 2.7) to demonstrate the results, the authors should examine more devices.

Author Response

The authors would like to thank the reviewers for their comprehensive and insightful review.  We are confident that responding to these comments will improve the quality and applicability of our paper.  We appreciate the opportunity to improve the clarity of our research.

  1. Formatting has been reviewed and several inconsistencies in labeling have been corrected.
  2. I believe the reviewer is referring to the statement that we “intended” to use three groups of Q values with the Pt Black, but that we were unable to fabricate values above 25 mC. This is an unnecessary piece of information.  The text has been modified to describe the groups of electrodes as “low” (<20mC) and “high” (> 20mC) to avoid this confusion.  In addition, the intention of the “low” and “high” designation was to ensure that we had an equal distribution of Q values in both the Nafion and non-Nafion coated groupings, not to compare specific Q values.  We hope that the reviewer agrees that this modification in categories produces groups that are adequate in size for the scope of this paper.

Round 2

Reviewer 2 Report

Dear Authors,

thank you for clarifying and making corrections to the text.

Regarding the mechanical properties, however, I still can't find an exact explanation - You wrote in response “Mechanical stability” has been defined in section 2.4 when describing Nafion..." First, I guess it is section 2.6, second, does mechanical stability mean adhesion? You provided references, but in your research work, since you write about changing mechanical properties, you should analyze them.

Author Response

We would like to thank the reviewer for the clarification on the ambiguity of the term “mechanical stability”.  The suggestion of “adhesion” is the correct interpretation of our intent.  The term “mechanical stability” has been replaced by “adhesion” throughout the document and a brief statement was added to justify the use of the Q value as a metric for evaluating adhesion.

Round 3

Reviewer 2 Report

Dear authors,

As a rule, if any parameters of the obtained samples are presented, they should be described in such a way that the reader can know what properties the given sample has. Anyway, this paper focuses on other parameters, so I believe that the mechanical properties (adhesion) have been sufficiently described.